# Rapid Epidemiological Analysis of Comorbidities and Treatments as risk factors for COVID-19 in Scotland (REACT-SCOT): A population-based case-control study

Paul M. McKeigue[1,2]*, Amanda Weir[2], Jen Bishop[2], Stuart J. McGurnaghan[3], Sharon Kennedy[4], David McAllister[2,5], Chris Robertson[6], Rachael Wood[4], Nazir Lone[1], Janet Murray[2], Thomas M. Caparrotta[3], Alison Smith-Palmer[2], David Goldberg[2], Jim McMenamin[2], Colin Ramsay[2], Sharon Hutchinson[2,7], Helen M. Colhoun[2,3], on behalf of Public Health Scotland COVID-19 Health Protection Study Group¶

1 Usher Institute, College of Medicine and Veterinary Medicine, University of Edinburgh, Edinburgh, Scotland, 2 Public Health Scotland, Glasgow, Scotland, 3 Institute of Genetics and Molecular Medicine, College of Medicine and Veterinary Medicine, University of Edinburgh, Edinburgh, Scotland, 4 NHS Information Services Division (Public Health Scotland), Edinburgh, Scotland, 5 Institute of Health and Wellbeing, University of Glasgow, Glasgow, Scotland, 6 Department of Mathematics and Statistics, University of Strathclyde, Glasgow, Scotland, 7 School of Health and Life Sciences, Glasgow Caledonian University, Glasgow, Scotland

¶ Membership of Public Health Scotland COVID-19 Health Protection Study Group is provided in the Acknowledgements.
* paul.mckeigue@ed.ac.uk

**Data Availability Statement:** The component datasets used here are available via the Public

## Abstract

### Background

The objectives of this study were to identify risk factors for severe coronavirus disease 2019 (COVID-19) and to lay the basis for risk stratification based on demographic data and health records.

### Methods and findings

The design was a matched case-control study. Severe COVID-19 was defined as either a positive nucleic acid test for severe acute respiratory syndrome coronavirus 2 (SARS-CoV-2) in the national database followed by entry to a critical care unit or death within 28 days or a death certificate with COVID-19 as underlying cause. Up to 10 controls per case matched for sex, age, and primary care practice were selected from the national population register. For this analysis—based on ascertainment of positive test results up to 6 June 2020, entry to critical care up to 14 June 2020, and deaths registered up to 14 June 2020—there were 36,948 controls and 4,272 cases, of which 1,894 (44%) were care home residents. All diagnostic codes from the past 5 years of hospitalisation records and all drug codes from prescriptions dispensed during the past 240 days were extracted. Rate ratios for severe COVID-19 were estimated by conditional logistic regression. In a logistic regression using the age-sex distribution of the national population, the odds ratios for severe disease were 2.87 for a 10-year

Benefits Privacy Panel for Health at https://www.informationgovernance.scot.nhs.uk/pbpphsc/ for researchers who meet the criteria for access to confidential data. All source code used for derivation of variables, statistical analysis and generation of this manuscript is available on https://github.com/pmckeigue/covid-scotland_public.

**Funding:** The author(s) received no specific funding for this work.

**Competing interests:** I have read the journal's policy and the authors of this manuscript have the following competing interests:HC receives research support and honoraria and is a member of advisory panels or speaker bureaus for Sanofi Aventis, Regeneron, Novartis, Novo-Nordisk and Eli Lilly. HC receives or has recently received non-binding research support from AstraZeneca and Novo-Nordisk. SH received honoraria from Gilead. TMC is a Diabetes UK 'Sir George Alberti Clinical Research Fellow' (Grant number: 18/0005786), although the views represented in this article are his own and not those of Diabetes UK. CR reports grants from Public Health Scotland, grants from MRC, during the conduct of the study; and Member of Chief Medical Officer of Scotland Scientific Advisory Group for COVID19 Member of SPI-M a subgroup of the UK Scientific Advisory Group for Epidemics Member of MHRA Advisory Group for Vaccine Safety. All other co-authors declare that no competing interest exists.

**Abbreviations:** BNF, British National Formulary; CHI, Community Health Index; COVID-19, coronavirus disease 2019; ECOSS, Electronic Communication of Surveillance in Scotland; ENCEPP, European Network of Centres for Pharmacoepidemiology and Pharmacovigilance; ICD-10, International Statistical Classification of Diseases Tenth Revision; NHS, National Health Service; ROC, receiver operator characteristic; SARS-CoV-2, severe acute respiratory syndrome coronavirus 2; SICSAG, Scottish Intensive Care Society and Audit Group; SIMD, Scottish Index of Multiple Deprivation; SMR, Scottish Morbidity Record; STROBE, Strengthening the Reporting of Observational Studies in Epidemiology.

increase in age and 1.63 for male sex. In the case-control analysis, the strongest risk factor was residence in a care home, with rate ratio 21.4 (95% CI 19.1–23.9, $p = 8 \times 10^{-644}$). Univariate rate ratios for conditions listed by public health agencies as conferring high risk were 2.75 (95% CI 1.96–3.88, $p = 6 \times 10^{-9}$) for type 1 diabetes, 1.60 (95% CI 1.48–1.74, $p = 8 \times 10^{-30}$) for type 2 diabetes, 1.49 (95% CI 1.37–1.61, $p = 3 \times 10^{-21}$) for ischemic heart disease, 2.23 (95% CI 2.08–2.39, $p = 4 \times 10^{-109}$) for other heart disease, 1.96 (95% CI 1.83–2.10, $p = 2 \times 10^{-78}$) for chronic lower respiratory tract disease, 4.06 (95% CI 3.15–5.23, $p = 3 \times 10^{-27}$) for chronic kidney disease, 5.4 (95% CI 4.9–5.8, $p = 1 \times 10^{-354}$) for neurological disease, 3.61 (95% CI 2.60–5.00, $p = 2 \times 10^{-14}$) for chronic liver disease, and 2.66 (95% CI 1.86–3.79, $p = 7 \times 10^{-8}$) for immune deficiency or suppression. Seventy-eight percent of cases and 52% of controls had at least one listed condition (51% of cases and 11% of controls under age 40). Severe disease was associated with encashment of at least one prescription in the past 9 months and with at least one hospital admission in the past 5 years (rate ratios 3.10 [95% CI 2.59–3.71] and 2.75 [95% CI 2.53–2.99], respectively) even after adjusting for the listed conditions. In those without listed conditions, significant associations with severe disease were seen across many hospital diagnoses and drug categories. Age and sex provided 2.58 bits of information for discrimination. A model based on demographic variables, listed conditions, hospital diagnoses, and prescriptions provided an additional 1.07 bits (C-statistic 0.804). A limitation of this study is that records from primary care were not available.

## Conclusions

We have shown that, along with older age and male sex, severe COVID-19 is strongly associated with past medical history across all age groups. Many comorbidities beyond the risk conditions designated by public health agencies contribute to this. A risk classifier that uses all the information available in health records, rather than only a limited set of conditions, will more accurately discriminate between low-risk and high-risk individuals who may require shielding until the epidemic is over.

## Author summary

### Why was this study done?

- Most people infected with severe acute respiratory syndrome coronavirus 2 (SARS-CoV-2) do not become seriously ill: risk of severe or fatal disease is associated with older age, male sex, and conditions designated by public health agencies, including asthma, diabetes, and heart disease.

- Studies reported so far have focused on these "listed conditions" but have not examined medical records systematically to identify possible risk factors for severe coronavirus disease 2019 (COVID-19).

- The objectives of this study were to identify risk factors for severe COVID-19 and to lay the basis for risk stratification based on electronic health records.

## What did the researchers do and find?

- Using Scotland's capability for linking electronic health records, we report the first systematic study of the relationship of severe or fatal COVID-19 to preexisting health conditions and other risk factors.

- Residents in care homes were 21 times more likely to develop severe disease than people of the same age and sex not living in care homes.

- The conditions associated with increased risk include not only those already designated by public health agencies—asthma, diabetes, heart disease, disabling neurological disease, kidney disease—but other diagnoses that are associated with frailty and poor health such as strokes and a history of falls.

- In those without any listed conditions, use of prescribed drugs acting on the digestive system or nervous system is associated with increased risk of severe COVID-19.

## What do these findings mean?

- The risk to younger individuals without any recent history of hospital admission or use of prescription drugs is very low.

- This study lays a basis for calculating a risk score based on electronic health records for every individual in the population and using it to advise those at high risk of severe disease to shield themselves when there is a COVID-19 epidemic in their locality.

## Background

Case series from many countries have suggested that, in those with severe coronavirus disease 2019 (COVID-19), the prevalence of diabetes and cardiovascular disease is higher than expected. For example, in a large United Kingdom series, the commonest comorbidities were cardiac disease, diabetes, chronic pulmonary disease, and asthma [1]. However, there are also anecdotal reports of apparently healthy young persons succumbing to disease [2].

Quantification of the risk associated with characteristics and comorbidities has been limited by the lack of comparisons with the background population [3–5]. Two recent studies in the UK have included population comparators and have reported associations of hospitalization with COVID-19 or death from COVID-19 with comorbidities including diabetes, asthma, and heart disease [6,7]. These studies have focused on conditions presumptively listed by public health agencies as increasing risk for COVID-19 based on case series data.

Here, we examine the frequency of sociodemographic factors and these listed conditions in all people with severe COVID-19 disease in Scotland compared to matched controls from the general population. In those without listed conditions, we report a systematic examination of the hospitalisation record and prescribing history in severe COVID-19 cases compared to controls. The objectives were to identify risk factors for severe COVID-19 and to lay the basis for risk stratification based on a predictive model.

## Methods

The protocol of the study dated 16 April 2020, together with all code used to prepare this manuscript, is available in a public repository (https://github.com/pmckeigue/covid-scotland_public). We modified the original protocol to align the list of risk conditions to be consistent with those designated by public health agencies and extended the list of drug classes under study to include all drugs. The study was registered with the European Network of Centres for Pharmacoepidemiology and Pharmacovigilance (ENCEPP number http://www.encepp.eu/encepp/viewResource.htm?id=35559EUPAS35558).

### Ethics statement

All record linkage studies using National Health Service (NHS) data in Scotland are governed by the Public Benefit and Privacy Panel for Health and Social Care, which includes patient and public representatives. Identifiable data were extracted by the Public Health Scotland Community Health Index (CHI) database and linkage team and de-identified before provision to the analysis team.

### Case definition and selection of matched controls

All individuals testing positive for nucleic acid for severe acute respiratory syndrome coronavirus 2 (SARS-CoV-2) were ascertained through the Electronic Communication of Surveillance in Scotland (ECOSS) database, which captures all virology testing in all NHS laboratories nationally. Linkage to other datasets was carried out using the CHI number, a unique identifier used in all care systems in Scotland. Admissions to critical care were obtained from the Scottish Intensive Care Society and Audit Group (SICSAG) database that captures admission to all critical care (intensive care or high-dependency) units and has returned a daily census of patients in critical care from the beginning of the COVID-19 epidemic. Death registrations were obtained from linkage to the National Register of Scotland. Severe or fatal COVID-19 was defined by either (1) a positive nucleic acid test followed by entry to critical care or death within 28 days or (2) a death certificate with COVID-19 as underlying cause. Using this definition ensures ascertainment of all severe cases even if they die without testing positive or entering critical care, whatever selection policies may have limited entry to critical care.

For each case, the CHI database was used to select up to 10 controls who were matched for sex and 1-year age band, were registered with the same primary care practice, and were alive and resident in Scotland on the same day as the first date that the case tested positive. For fatal cases who had not tested positive, the incident date was assigned as 14 days before death. To ensure that cases and controls were representative of the same population at risk, the 0.6% of cases that were recorded on the CHI database as no longer alive and resident in Scotland on the day that ECOSS recorded them as testing positive were also excluded. As this is an incidence density sampling design, it is possible and correct for an individual to appear in the dataset more than once, initially as a control and subsequently as a case.

For this analysis based on ascertainment of positive test results up to 6 June 2020, entry to critical care up to 14 June 2020, and deaths registered up to 14 June 2020, there were 4,272 cases and 36,948 controls. Among fatal cases, 94% of deaths were registered within 5 days.

### Demographic data

Residence in a care home was ascertained from the CHI database. Socioeconomic status was assigned as the quintile of the Scottish Index of Multiple Deprivation (SIMD), an indicator based on postal code. For 74% of controls and 85% of cases, self-assigned ethnicity of the

individual—based on the categories used in Scotland's Census—had been recorded in the Scottish Morbidity Record (SMR).

## Morbidity and drug prescribing

For all cases and controls, International Statistical Classification of Diseases Tenth Revision (ICD-10) diagnostic codes were extracted from the last 5 years of hospital discharge records in the SMR (SMR01), excluding records of discharges less than 25 days before testing positive for SARS-CoV-2 and using all codes on the discharge. Diagnostic coding under ICD Chapters V (Mental, Behavioural and Neurodevelopmental) and XV (Pregnancy) is incomplete because most psychiatric and maternity unit returns are not captured in SMR01. British National Formulary (BNF) drug codes for dispensed prescriptions issued in primary care were extracted from the Scottish Prescribing Information System [8]. A cutoff date of 15 days before the incident date (date of testing positive for SARS-CoV-2, or 14 days before death for fatal cases without a positive test) was set, and prescriptions dispensed during a 240-day interval before this cutoff date were included. For this analysis, prescription codes from BNF chapters 14 and above—comprising dressings, appliances, vaccines, anaesthesia, and other preparations—were grouped as "Other."

We began by scoring a specific list of conditions that have been designated as risk conditions for COVID-19 by public health agencies [9]. A separate list of conditions designates "clinically extremely vulnerable" individuals who were advised to shield themselves completely since 24 March 2020: this list includes solid organ transplant recipients, people receiving chemotherapy for cancer, and people with cystic fibrosis or leukaemia. We did not separately tabulate these conditions because we expected these individuals to be underrepresented among cases if shielding was adequate.

The 8 listed conditions were scored based on diagnostic codes in any hospital discharge record during the last 5 years, or encashed prescription of a drug for which the only indications are in that group of diagnostic codes. The R script included as Supporting Information contains the derivations of these variables from ICD-10 codes and BNF drug codes. Diagnosed cases of diabetes were identified through linkage to the national diabetes register (SCI-Diabetes), with a clinical classification of diabetes type as type 1, type 2, or Other/Unknown.

## Statistical methods

To estimate the relationship of cumulative incidence and mortality from COVID-19 to age and sex, logistic regression models were fitted to the proportions of cases and noncases in the Scottish population, using the estimated population of Scotland in mid-year 2019, which was available by 1-year age group up to age 90 years. To allow for possible nonlinearity of the relationship of the logit of risk to age, we also fitted generalized additive models, implemented in the R function gam::gam, with default smoothing function.

For the case-control study, all estimates of associations with severe COVID-19 were based on conditional logistic regression, implemented as Cox regression in the R function survival::clogit [10]. Among those cases and controls without any of the predefined conditions, we then further examined associations of ICD-10 and BNF chapter with severe COVID-19. Restriction of cases and controls, for instance, to exclude those with any listed condition may generate strata that do not contain at least one case and at least one control, but these strata are ignored by the conditional logistic regression model as they do not contribute to the conditional likelihood. With incidence density sampling, the odds ratios in conditional logistic regression models are equivalent to rate ratios. Note that odds ratios in a matched case-control study are based on the conditional likelihood; the unconditional odds ratios calculable from the

frequencies of exposure in cases and controls will differ from these and are biased towards the null [11]. Although matching on primary care practice will match to some extent for associated variables such as care home residence, socioeconomic disadvantage, and prescribing practice, the effects of these variables are still estimated correctly by the conditional odds ratios but with less precision than in an unmatched study of the same size [11].

To construct risk prediction models, we used stepwise regression alternating between forward and backward steps to maximize the AIC, implemented in the R function stats::step. To evaluate the contribution of the listed conditions to risk prediction, and the incremental contribution of other information in hospitalisation and prescription records after assigning these conditions, predictive models were constructed from 3 sets of variables: a baseline set consisting only of demographic variables; a set that included indicator variables for each listed condition; and an extended set that included demographic variables, indicator variables for listed conditions, and indicator variables for hospital diagnoses in each ICD-10 chapter and prescriptions in each BNF chapter.

The performance of the risk prediction model in classifying cases versus noncases of severe COVID-19 was examined by 10-fold cross-validation. We calculated the performance calculated over all test folds using the C-statistic but also using the "expected information for discrimination" $\Lambda$ expressed in bits [12]. The use of bits (logarithms to base 2) to quantify information is standard in information theory: one bit can be defined as the quantity of information that halves the hypothesis space. Although readers may be unfamiliar with the expected information for discrimination $\Lambda$, it has several properties that make it more useful than the C-statistic for quantifying increments in the performance of a risk prediction model [12]. A key advantage of using $\Lambda$ is that contributions of independent predictors can be added. Therefore, in this study we can add the predictive information from a logistic model of age and sex in the general population to the predictive information provided by other risk factors from the case-control study matched for age and sex.

## Results

### Incidence and mortality from severe COVID-19 in the Scottish population

Fig 1 shows the relationships of incidence and mortality rates to age for each sex separately. The relationship of mortality to age is almost exactly linear on a logit scale, and the lines for male and female mortality are almost parallel. In models that included age and sex as covariates, the odds ratio associated with a 10-year increase in age was 2.87 for all severe disease and 3.7 for fatal disease. The odds ratio associated with male sex was 1.63 for all severe disease and 1.58 for fatal disease. For severe cases as defined in this study, the sex differential is narrow up to about age 50 but widens between ages 50 and 70 years. Therefore, at younger ages, the ratio of critical care admissions to total fatalities is higher in women than in men, but at later ages, the ratio of critical admissions to total fatalities is higher in men.

### Risk factors

**Sociodemographic factors.** Table 1 shows univariate associations of demographic factors with severe disease. Residence in a care home was by far the strongest risk factor for severe disease. Higher risk of severe disease was also associated with socioeconomic deprivation. In the 85% of cases and 74% of controls for whom ethnicity of the individual had been recorded in the SMR, there were few nonwhite individuals, and the confidence limits for the rate ratios by ethnic group were wide.

**Factors derived from hospitalisation and prescribing records.** Prevalence of the listed conditions in cases and controls by age band is shown in Table 2. Twenty-nine (51%) of the

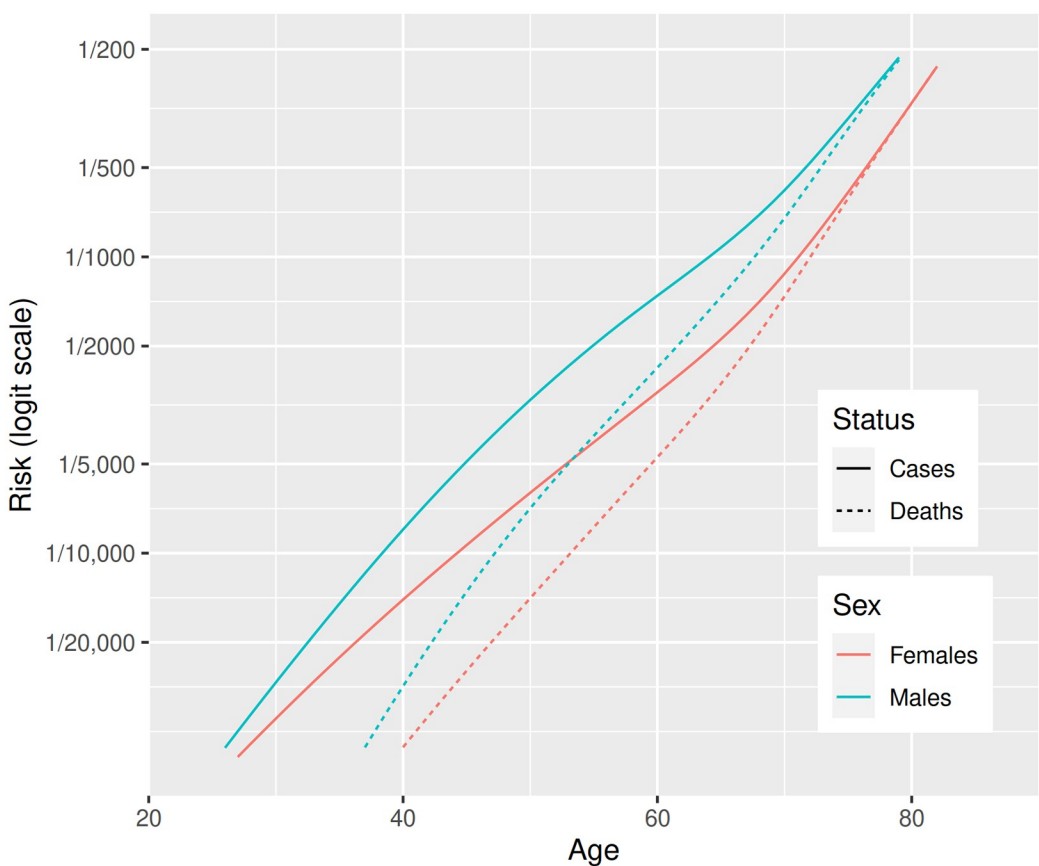

**Fig 1. Incidence of severe and fatal COVID-19 in Scotland by age and sex: Generalized additive models fitted to severe and fatal cases for males and females separately.** COVID-19, coronavirus disease 2019.

**Table 1. Univariate associations of severe disease with demographic factors.**

|  | Controls | Cases | Rate ratio (95% CI) | *p* -Value |
|---|---|---|---|---|
| **Number of individuals (entire sample)** | 36,948 | 4,272 |  |  |
| **Most deprived SIMD quintile** | 8,559 (23%) | 1,121 (26%) |  |  |
|  | 7,956 (22%) | 935 (22%) | 0.86 (0.78–0.95) | 0.003 |
|  | 6,730 (18%) | 826 (19%) | 0.88 (0.79–0.98) | 0.02 |
|  | 6,558 (18%) | 773 (18%) | 0.81 (0.72–0.90) | $2 \times 10^{-4}$ |
| **Least deprived SIMD quintile** | 7,119 (19%) | 614 (14%) | 0.54 (0.48–0.62) | $4 \times 10^{-21}$ |
| **Care home** | 2,935 (8%) | 1,894 (44%) | 21.4 (19.1–23.9) | $8 \times 10^{-644}$ |
| **Number of individuals (with SMR record of ethnicity)** | 27,230 | 3,648 |  |  |
| **White** | 26,908 (99%) | 3,596 (99%) |  |  |
| **South Asian** | 145 (1%) | 27 (1%) | 1.26 (0.81–1.97) | 0.3 |
| **Black** | 35 (0%) | 5 (0%) | 1.16 (0.44–3.04) | 0.8 |
| **Other** | 142 (1%) | 20 (1%) | 1.01 (0.63–1.65) | 1 |

**Abbreviations**: SIMD, Scottish Index of Multiple Deprivation; SMR, Scottish Morbidity Record

**Table 2. Frequencies of risk factors in cases and controls, by age group.**

|  | 0–39 years | | 40–59 years | | 60–74 years | | 75+ years | |
| --- | --- | --- | --- | --- | --- | --- | --- | --- |
|  | Controls (570) | Cases (57) | Controls (4,168) | Cases (418) | Controls (8,734) | Cases (881) | Controls (23,476) | Cases (2,916) |
| Care home | 0 (0%) | 1 (2%) | 4 (0%) | 19 (5%) | 90 (1%) | 176 (20%) | 2,841 (12%) | 1,698 (58%) |
| Any prescription | 311 (55%) | 47 (82%) | 2,916 (70%) | 371 (89%) | 7,591 (87%) | 839 (95%) | 22,665 (97%) | 2,872 (98%) |
| Any admission | 143 (25%) | 26 (46%) | 1,474 (35%) | 249 (60%) | 4,394 (50%) | 674 (77%) | 16,527 (70%) | 2,508 (86%) |
| Any listed condition | 64 (11%) | 29 (51%) | 1,028 (25%) | 225 (54%) | 3,764 (43%) | 637 (72%) | 14,299 (61%) | 2,436 (84%) |
| Diagnosis or prescription | 344 (60%) | 48 (84%) | 3,115 (75%) | 386 (92%) | 7,815 (89%) | 859 (98%) | 22,894 (98%) | 2,901 (99%) |
| Type 1 diabetes | 0 (0%) | 2 (4%) | 46 (1%) | 12 (3%) | 42 (0%) | 8 (1%) | 70 (0%) | 21 (1%) |
| Type 2 diabetes | 3 (1%) | 2 (4%) | 250 (6%) | 75 (18%) | 1,319 (15%) | 219 (25%) | 3,970 (17%) | 613 (21%) |
| Other/unknown type | 2 (0%) | 4 (7%) | 24 (1%) | 14 (3%) | 73 (1%) | 8 (1%) | 184 (1%) | 22 (1%) |
| Ischaemic heart disease | 2 (0%) | 0 (0%) | 126 (3%) | 34 (8%) | 955 (11%) | 170 (19%) | 4,392 (19%) | 702 (24%) |
| Other heart disease | 6 (1%) | 7 (12%) | 176 (4%) | 66 (16%) | 1,236 (14%) | 265 (30%) | 7,192 (31%) | 1,411 (48%) |
| Asthma or chronic airway disease | 49 (9%) | 22 (39%) | 567 (14%) | 114 (27%) | 1,686 (19%) | 328 (37%) | 5,306 (23%) | 970 (33%) |
| Chronic kidney disease or transplant recipient | 1 (0%) | 0 (0%) | 8 (0%) | 16 (4%) | 30 (0%) | 24 (3%) | 163 (1%) | 57 (2%) |
| Neurological (except epilepsy) or dementia | 3 (1%) | 7 (12%) | 61 (1%) | 43 (10%) | 321 (4%) | 177 (20%) | 2,897 (12%) | 1,154 (40%) |
| Liver disease | 1 (0%) | 0 (0%) | 20 (0%) | 10 (2%) | 53 (1%) | 21 (2%) | 59 (0%) | 20 (1%) |
| Immune deficiency or suppression | 2 (0%) | 1 (2%) | 18 (0%) | 13 (3%) | 47 (1%) | 15 (2%) | 76 (0%) | 11 (0%) |

cases aged under 40 years had at least one listed condition, compared with only 64 (11%) of the controls. In those aged 75+ years, 2,346 (84%) of the cases and 14,299 (61%) of the controls had at least one listed condition. Among those aged under 40 years, 48 (84%) of the cases and 344 (60%) of the controls had either a hospital admission in the last 5 years or a dispensed prescription in the last 240 days. Differences in prescription rates between cases and controls narrowed with increasing age.

Across all age groups, 3,327 (78%) of severe cases and 19,155 (52%) of controls had at least one of the listed conditions. As shown in Table 3, all the listed conditions were more frequent in cases than controls except for immune conditions in the 75+ age group. The rate ratio

**Table 3. Associations of severe disease with listed conditions over all age groups.**

|  |  |  | Univariate | | Multivariable | |
| --- | --- | --- | --- | --- | --- | --- |
|  | Controls (36,948) | Cases (4,272) | Rate ratio (95% CI) | p-Value | Rate ratio (95% CI) | p-Value |
| Care home | 2,935 (8%) | 1,894 (44%) | 21.4 (19.1–23.9) | $8 \times 10^{-644}$ | 14.7 (13.1–16.6) | $1 \times 10^{-431}$ |
| Any prescription | 33,483 (91%) | 4,129 (97%) | 3.10 (2.59–3.71) | $8 \times 10^{-35}$ | 1.83 (1.51–2.22) | $6 \times 10^{-10}$ |
| Any admission | 22,538 (61%) | 3,457 (81%) | 2.75 (2.53–2.99) | $2 \times 10^{-124}$ | 1.56 (1.41–1.72) | $1 \times 10^{-18}$ |
| Type 1 diabetes | 158 (0%) | 43 (1%) | 2.75 (1.96–3.88) | $6 \times 10^{-9}$ | 1.56 (1.05–2.32) | 0.03 |
| Type 2 diabetes | 5,542 (15%) | 909 (21%) | 1.60 (1.48–1.74) | $8 \times 10^{-30}$ | 1.42 (1.29–1.56) | $3 \times 10^{-13}$ |
| Other/unknown type | 283 (1%) | 48 (1%) | 1.74 (1.28–2.38) | $4 \times 10^{-4}$ | 1.58 (1.11–2.27) | 0.01 |
| Ischaemic heart disease | 5,475 (15%) | 906 (21%) | 1.49 (1.37–1.61) | $3 \times 10^{-21}$ | 1.08 (0.98–1.20) | 0.1 |
| Other heart disease | 8,610 (23%) | 1,749 (41%) | 2.23 (2.08–2.39) | $4 \times 10^{-109}$ | 1.33 (1.22–1.46) | $2 \times 10^{-10}$ |
| Asthma or chronic airway disease | 7,608 (21%) | 1,434 (34%) | 1.96 (1.83–2.10) | $2 \times 10^{-78}$ | 1.54 (1.42–1.68) | $7 \times 10^{-25}$ |
| Chronic kidney disease or transplant recipient | 202 (1%) | 97 (2%) | 4.06 (3.15–5.23) | $3 \times 10^{-27}$ | 2.88 (2.13–3.89) | $7 \times 10^{-12}$ |
| Neurological (except epilepsy) or dementia | 3,282 (9%) | 1,381 (32%) | 5.4 (4.9–5.8) | $1 \times 10^{-354}$ | 2.00 (1.81–2.21) | $2 \times 10^{-42}$ |
| Liver disease | 133 (0%) | 51 (1%) | 3.61 (2.60–5.00) | $2 \times 10^{-14}$ | 1.93 (1.32–2.81) | $6 \times 10^{-4}$ |
| Immune deficiency or suppression | 143 (0%) | 40 (1%) | 2.66 (1.86–3.79) | $7 \times 10^{-8}$ | 1.67 (1.10–2.52) | 0.01 |

associated with type 1 diabetes was higher than that for type 2 diabetes. The rate ratio was 1.49 (95% CI 1.37–1.61) for ischemic heart disease compared to 2.23 (95% CI 2.08–2.39) for the broad category "other heart disease." In multivariable analysis, ischemic heart disease was not independently associated with severity, whereas other heart disease remained strongly associated. In those without a listed condition, 873 (92%) of the cases and 15,052 (85%) of the controls had either a recent admission or a prescription. In those aged under 60 years without a listed condition, 184 (83%) of the cases and 2,376 (65%) of the controls had either a recent admission or a prescription.

S1–S3 Tables examine these associations by age group, with the 0–39 and 40–59 year age bands combined. All listed conditions were associated with severe disease in each age band. In those aged under 60 years, the rate ratio was 3.70 (95% CI 2.01–6.79) for type 1 diabetes and 3.70 (95% CI 2.80–4.90) for type 2 diabetes. The multivariable analyses shown in Table 3 and S1–S3 Tables show that, overall and in each age group, any admission to hospital in the past 5 years was strongly and independently associated with severe disease even after adjusting for care home residence and listed conditions. Dispensing of any prescription in the past year was associated with severe disease in multivariable analyses in the 2 younger age bands. Table 4 shows that, in each age group, the proportion of fatal cases who had not had either a hospital admission in the last 5 years or a dispensed prescription in the last year was very low.

## Comparison of fatal and nonfatal cases

S4 Table shows a breakdown of severe cases by test-positive status of the patient, entry to critical care, and fatal versus nonfatal outcome. Severe cases who entered critical care were much younger than severe cases never entering critical care. Most severe cases who were resident in a care home never entered critical care. Among fatal cases who did not enter critical care, the distribution of age and other risk factors was similar in those with and without a positive test result, except that the proportion of care home residents was higher among those without a positive test result. Among those entering critical care, median age, proportion of males, and prevalence or prior comorbidities were higher in fatal than in nonfatal cases.

## Systematic analysis of diagnoses associated with severe disease

The association of severe COVID-19 with prior hospital admission was examined further by testing for association of hospitalisations at each ICD-10 chapter level with severe COVID-19, among those without any of the listed conditions. These results are shown in S5 Table. In univariate analyses, almost all ICD-10 chapters, with the exception of Chapters VII (eye), VIII

**Table 4. Proportions of fatal cases and matched controls without and with a dispensed prescription or hospital diagnosis, by age group.**

|  | Controls | Fatal cases |
|---|---|---|
| **Age <60 years** | | |
| No prescription or diagnosis | 1,305 (26%) | 15 (7%) |
| Prescription or diagnosis | 3,696 (74%) | 197 (93%) |
| **Age 60–74 years** | | |
| No prescription or diagnosis | 929 (10%) | 12 (2%) |
| Prescription or diagnosis | 7,994 (90%) | 680 (98%) |
| **Age 75+** | | |
| No prescription or diagnosis | 583 (2%) | 14 (0%) |
| Prescription or diagnosis | 22,924 (98%) | 2,871 (100%) |

(ear), and XV (pregnancy), were associated with increased risk of severe disease. In a multivariable analysis, the strongest associations were with diagnoses in ICD chapters IV (mental disorders) and X (respiratory). S7 Table extracts univariate associations with ICD-10 subchapters in those without any listed conditions. This table is filtered to show only subchapters for which there are at least 50 cases and controls and the univariate $p$-value is <0.001. This shows that many subchapter diagnoses are associated with markedly higher risk of severe COVID-19.

### Associations of prescribed drugs with severe disease

As shown in Table 3 and S1–S3 Tables, encashment of at least one prescription in the last year was associated with severe disease. The univariate rate ratio associated with this variable varies from 3.74 (95% CI 2.79–5.01) in those aged under 60 years to 2.30 (95% CI 1.69–3.14) in those aged 75 years and over. In a multivariable analysis adjusting for care home residence, any hospital admission, and listed conditions, these rate ratios were reduced to 2.12 (95% CI 1.55–2.90) and 1.13 (95% CI 0.80–1.60), respectively.

To investigate this further, we partitioned the "Any prescription" variable into indicator variables for each chapter of the BNF, in which drugs are grouped by broad indication, and restricted the analysis to those without one of the listed conditions. S6 Table shows these associations. In univariate analyses, prescriptions in almost all BNF chapters were associated with severe disease. In a multivariable analysis of all chapters, the strongest independent associations with severe disease were with prescriptions in chapters 1 (gastrointestinal), 4 (central nervous system), 5 (infections), 9 (nutrition and blood), and 14+ (other, mostly dressings and appliances).

### Construction of a multivariable risk prediction model

The variables retained from the extended variable set (demographic variables, listed conditions, hospital diagnoses in each ICD-10 chapter, prescriptions in each BNF chapter) are shown in S8 Table. Coefficients for specific conditions here should not be interpreted as effect estimates, as global variables for any hospital diagnosis and any listed condition have been included in the model. The predictive performance of the model chosen by stepwise regression was estimated by 10-fold cross-validation. Observed and predicted case-control status was compared within each stratum over all test folds. Table 5 shows that, in comparison with using only demographic variables and listed conditions, using the extended variable set increased the C-statistic from 0.776 to 0.804 and the expected information for discrimination $\Lambda$ from 0.88 bits to 1.07 bits.

Fig 2 shows the distributions in cases and controls of the weight of evidence favouring case over control status from the model based on the extended variable set with a footnote explaining how $\Lambda$ is derived. This shows, as expected for a multifactorial classifier, that the distribution in controls is approximately Gaussian: there is no clear divide between high-risk and low-risk individuals of the same age and sex. The distribution in cases is bimodal; the second mode

**Table 5. Prediction of severe COVID-19: Cross-validation of models chosen by stepwise regression.**

|  | Cases/controls | Crude C-statistic | Adjusted C-statistic | Crude$\Lambda$ (bits) | Adjusted$\Lambda$ (bits) | Test log-likelihood (nats) |
|---|---|---|---|---|---|---|
| **Demographic only** | 2,724/19,509 | 0.737 | 0.716 | 0.65 | 0.58 | 0.0 |
| **Demographic + listed conditions** | 2,724/19,509 | 0.794 | 0.776 | 0.95 | 0.88 | 389.8 |
| **Extended variable set** | 2,724/19,509 | 0.812 | 0.804 | 1.11 | 1.07 | 596.7 |

Abbreviation: COVID-19, coronavirus disease 2019

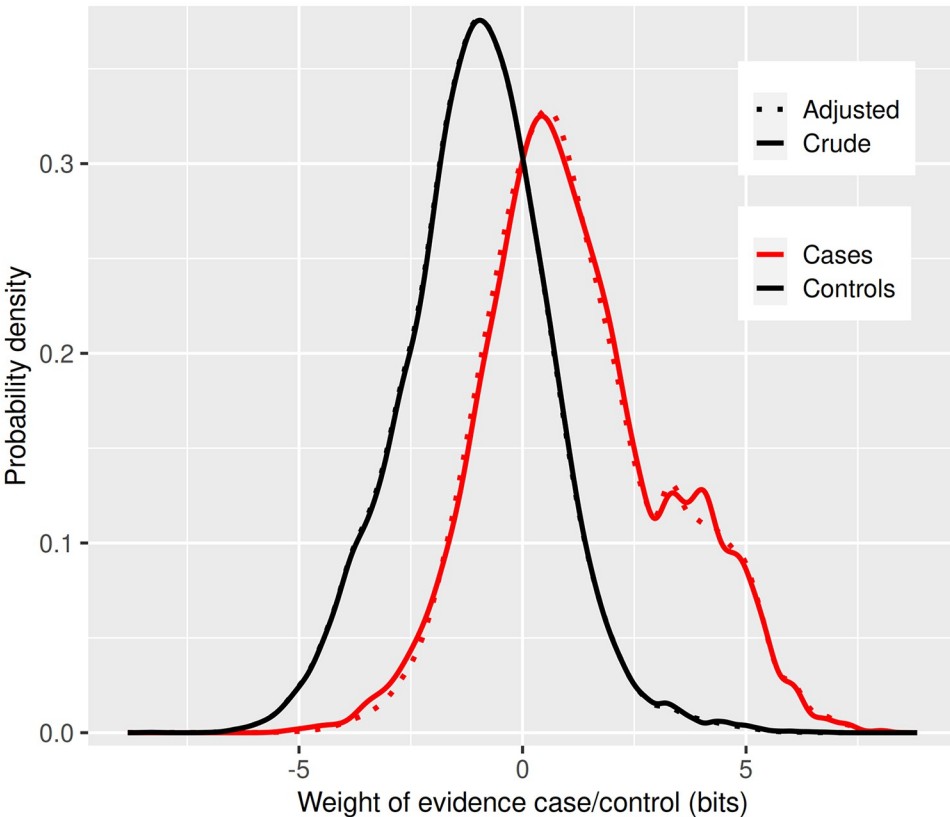

**Fig 2. Cross-validation of model chosen by stepwise regression using extended variable set: Class-conditional distributions of weight of evidence.** For each individual, the risk prediction model outputs the posterior probability of case status, which can also be expressed as the posterior odds. Dividing the posterior odds by the prior odds gives the likelihood ratio favouring case over noncase status for an individual. The weight of evidence $W$ is the logarithm of this ratio. The distributions of $W$ in cases and controls in the test data are plotted in Fig 2. For a classifier, the further apart these curves are, the better the predictive performance. The expected information for discrimination $\Lambda$ is the average of the mean of the distribution of $W$ in cases and $-1$ times the mean of the distribution of $W$ in controls. The distributions have been adjusted by taking a weighted average to make them mathematically consistent [12].

of this distribution represents care home residents. Fig 3 shows the receiver operating characteristic curve with a footnote explaining its derivation from the distributions of the weights of evidence.

The information for discrimination obtained from the matched case-control study which conditions on age and sex (1.07 bits) can be added to the information for discrimination obtained from the logistic regression on age and sex in the population (2.58 bits). This gives 3.65 bits as the total information for discrimination of a risk classifier that would be obtained in the population.

## Discussion

### Sociodemographic factors

This analysis confirms that risk for severe COVID-19 is associated with increasing age, male sex, and socioeconomic deprivation. The slope of the relationship of severe disease (on the scale of log odds) to age is less steep than the slope of the relationship of fatal disease to age. Residence in a care home was associated with a 21-fold increased rate of severe COVID-19 in this age-matched analysis, reduced to 15-fold by adjustment for listed conditions. This excess

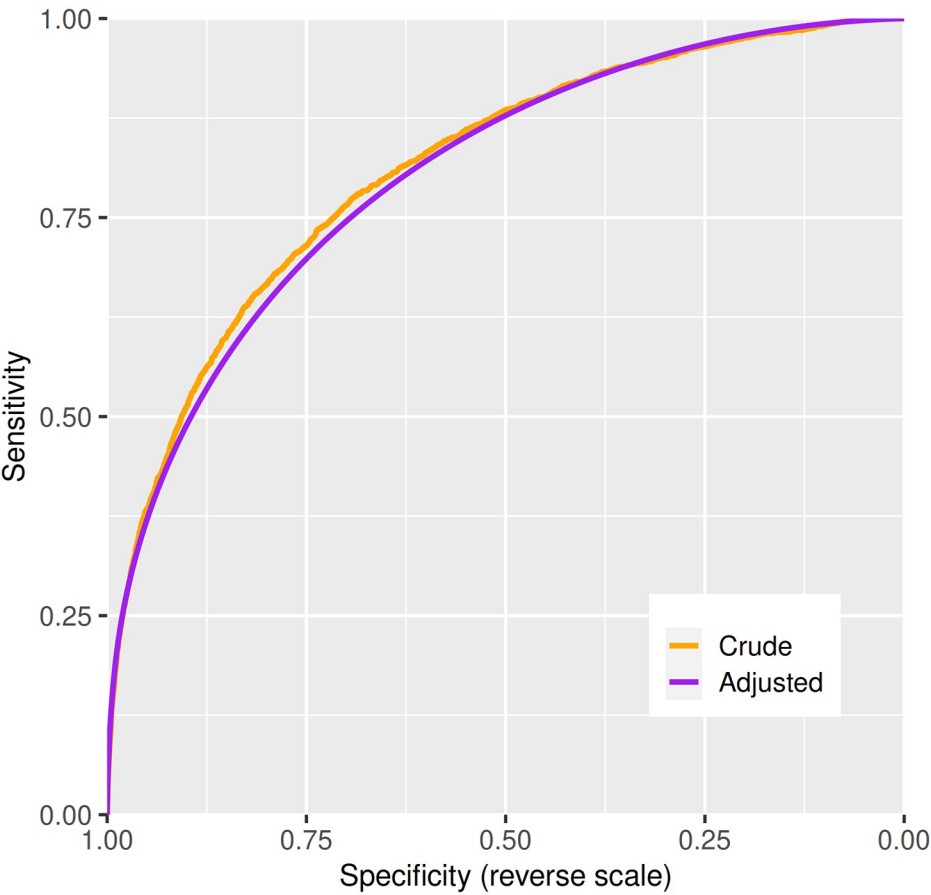

**Fig 3. Cross-validation of model chosen by stepwise regression using extended variable set: ROC curve.** The ROC curve is computed by calculating at each value of the risk score the sensitivity and specificity of a classifier that uses this value as the threshold for classifying cases and noncases. Using the adjusted distributions from Fig 2 gives a curve that is concave downwards. The C-statistic is the area under this curve, computed as the probability of correctly classifying a case/noncase pair using the risk score, evaluated over all possible such pairs in the dataset. ROC, receiver operator characteristic.

risk is likely to reflect both the spread of the epidemic in care homes and residual confounding by frailty.

As the proportion of the Scottish population that is of nonwhite ethnicity is low and the assignment of ethnicity in this dataset is incomplete, the confidence intervals for the rate ratios associated with South Asian and black ethnicity are wide. Studies from England [6,7,13] have reported elevations in risk of hospitalised and fatal COVID-19 in nonwhite ethnic groups; in the OpenSAFELY study, the risk ratios for fatal COVID-19 associated with black and Asian ethnicity were 1.7 and 1.6, respectively. The confidence intervals in this study are compatible with the effect sizes estimated in England.

## Comorbidities

We have confirmed that the moderate risk conditions designated by the NHS and other agencies [9] are associated with increased risk of severe COVID-19. The rate ratios of 2.8 for type 1 diabetes and 1.6 for type 2 diabetes are broadly similar to those reported in the UK Biobank [13] and OpenSAFELY [7] studies. We confirm the higher risk with asthma and chronic lung disease and liver disease reported in these and earlier studies. The rate ratios associated with

these risk conditions vary with age: for example, the rate ratio associated with diabetes is higher at younger ages. An unexpected finding was that the risk associated with other forms of heart disease is higher than that associated with ischaemic heart disease. This category includes conditions such as atrial fibrillation, cardiomyopathies, and heart failure. One of the highest rate ratios is that associated with chronic kidney disease. Prevention of nosocomial transmission in dialysis units may help to reduce this risk. Over all age groups, 78% of severe cases had at least one of these listed conditions. In this dataset, it is not possible to adequately examine the risk associated with neoplasms as we cannot separately identify those who were advised to shield themselves because they had active neoplasms of lymphoid or hematopoietic tissue or were receiving treatments that affect the immune system. We plan to explore this in a separate study based on linkage to records of shielding advice. In patients without any listed conditions, further systematic evaluation of past hospitalisation history did not reveal a sparse set of underlying conditions; instead, many diagnoses were associated with severe COVID-19.

Public health agencies [14] and media reports of apparently healthy young people succumbing to severe COVID-19 [2] have disseminated the message that all are at risk of severe COVID-19 whatever their age or health status. However, we found that half of cases who were under 40 years old had at least one of the listed conditions, and among those who did not have one of these conditions, the proportions who had at least one prior hospitalisation or dispensed prescription were higher in cases than in controls. In all age groups, very few of the fatal cases had not had either a hospital admission in the past 5 years or a dispensed prescription in the past year.

A striking finding of this study was the association of severe COVID-19 with dispensing of at least one prescription in the 240-day interval preceding the cutoff of 15 days before diagnosis, only partly explained by higher rates of prescribing among those with listed conditions. Partitioning of this association between BNF chapters, which represent broad indication-based drug classes, showed that prescribing of drugs for the gastrointestinal and central nervous systems, together with nutritional supplements, contributed to this association. Although it is likely that most associations of severe COVID-19 with drug prescribing are attributable to the indications for which these drugs were prescribed—or to more diffuse frailty, especially in older persons—causal effects of drugs or direct effects of polypharmacy on susceptibility cannot be ruled out. These associations are explored in a separate paper.

### Relevance to policy

As lockdown restrictions are eased, there is general agreement that vulnerable individuals will require shielding, even if the restart of the epidemic can be slowed or suppressed by mass testing, contact tracing, and isolation of those who test positive. The "stratify and shield" policy option [15]—in which high-risk individuals are shielded for a defined period while the epidemic is allowed to run relatively quickly in low-risk individuals until herd immunity is attained—depends critically on informative risk discrimination. So too does the similarly named "segment and shield" option [16], which has the opposite objective of keeping transmissions low. Although in this preliminary study we have not used the full repertoire of machine learning methods available for constructing predictive models, we have shown that a model based on health records provides 1.07 bits of information for discrimination conditional on age and sex. Adding this to the 2.58 bits provided by age and sex gives a total information for discrimination of 3.7 bits. We have shown elsewhere that this level of predictive performance would allow at least 80% of those at risk of severe or fatal disease to be allocated to a shielded group that composes no more than 15% of the population [15].

As awareness grows of how risk varies between individuals, individuals will seek information about their own level of risk. A key implication of our results is that risk of severe or fatal disease is multifactorial. The rate ratio of 2.9 associated with a 10-year increase in age is stronger than the rate ratios associated with common diseases such as asthma or type 2 diabetes that are listed as conditions associated with high risk. A corollary of this is that a crude classification based on assigning all persons with a listed condition to a group for whom shielding is recommended will have poor specificity, as one-quarter of those aged 60–74 years in the population have at least one of the listed conditions that we examined. It will also exclude many people at high risk because they have multiple risk factors each of small effect. A more meaningful way to score risk for an individual would be to use all available information to calculate a "COVID age" as the age at which the average risk for someone of the same sex in the population equates to the risk for the individual under study. Thus, the rate ratio of 2.8 associated with type 1 diabetes equates to an increase of 9.8 years in COVID age. In Scotland, it is technically possible to use existing electronic health records to calculate a risk score for every individual in the population, though more work would be required to develop this as a basis for official advice and individual decisions.

## Methodological strengths and weaknesses

Most reports of disease associations with COVID-19 have been case series. There have been few reports based on evaluating these associations in the population through cohort or case-control studies. With this matched case-control design using incidence density sampling, we have been able to estimate rate ratios conditional on age and sex. The OpenSAFELY study has explored associations of a similar set of risk conditions with in-hospital COVID-19 deaths [7] but has not yet reported a systematic evaluation of the rest of the medical record including prescription records. Although we have records of encashment of prescriptions, we do not at present have access to other primary care data, which would contain additional information on morbidity and measurements such as body mass index. A strength of our study, however, is that hospital discharge diagnoses are coded to ICD-10 by trained coders, in contrast to the coding systems used in primary care databases that do not map to recognized disease classifications. Associations with ethnicity and other sociodemographic factors are not necessarily generalizable from Scotland to other populations.

## Conclusion

This study confirms that risk of severe COVID-19 is associated with sociodemographic factors and with chronic conditions such as diabetes, asthma, circulatory disease, and others. However, the associations with preexisting disease are not just with a small set of conditions that contribute to risk but with many conditions as demonstrated by associations with past medical and prescribing history in relation to multiple physiological systems. As countries attempt to emerge from lockdown while protecting vulnerable individuals, multivariable classifiers rather than crude rule-based approaches will be needed to define those most at risk of developing severe disease.

## Supporting information

**S1 Table. Associations of severe disease with listed conditions in those aged less than 60.**
(PDF)

**S2 Table. Associations of severe disease with listed conditions in those aged 60–74 years.**
(PDF)

**S3 Table. Associations of severe disease with listed conditions in those aged 75 years and over.**
(PDF)

**S4 Table. Comparison of severe non-fatal and fatal cases, by test positive status and entry to critical care.**
(PDF)

**S5 Table. Associations of severe disease with hospital diagnoses by ICD chapter in last 5 years, in those without any listed condition.**
(PDF)

**S6 Table. Associations of severe disease with prescribed drugs by BNF chapter in those without any listed condition.**
(PDF)

**S7 Table. Univariate associations of severe disease with hospital diagnoses by ICD sub-chapters in those without any listed conditions: Rows retained are those with p < 0.001 and at least 50 cases and controls.**
(PDF)

**S8 Table. Stepwise regression: Variables retained in model for severe disease.**
(PDF)

**S1 RECORD Checklist.**
(DOCX)

**S1 STROBE Checklist. STROBE, Strengthening the Reporting of Observational Studies in Epidemiology.**
(DOCX)

## Acknowledgments

We thank all staff in critical care units who submitted data to the SICSAG database, the Scottish Morbidity Record Data Team, the staff of the National Register of Scotland, the Public Health Scotland (PHS) Terminology Services, the HPS COVID-19 Laboratory and Testing Cell, the NHS Scotland Diagnostic Virology Laboratories, and Nicola Rowan (PHS) for coordinating this collaboration.

### Public Health Scotland COVID-19 Health Protection Study Group

From Health Protection Scotland (Public Health Scotland), Meridian Court, 5 Cadogan Street, Glasgow G2 6QE, Scotland: Alice Whettlock, Allan McLeod, Andrew Gasiorowski, Andrew Merrick, Andy McAuley, April Went, Calum Purdie, Colin Fischbacher, Colin Ramsay, David Bailey, David Henderson, Diogo Marques, Eisin McDonald, Genna Drennan, Graeme Gowans, Graeme Reid, Heather Murdoch, Jade Carruthers, Janet Fleming, Jade Carruthers, Joseph Jasperse, Josie Murray, Karen Heatlie, Lindsay Mathie, Lorraine Donaldson, Martin Paton, Martin Reid, Melissa Llano, Michelle Murphy-Hall, Paul Smith, Ros Hall, Ross Cameron, and Susan Brownlie.

From NHS National Services Scotland, Meridian Court, 5 Cadogan Street, Glasgow G2 6QE, Scotland: Adam Gaffney, Aynsley Milne, Christopher Sullivan, Edward McArdle, Elaine Glass, Johanna Young, William Malcolm, and Jodie McCoubrey.

## Author Contributions

**Conceptualization:** Paul M. McKeigue, Amanda Weir, Jen Bishop, Sharon Kennedy, David McAllister, Rachael Wood, Janet Murray, David Goldberg, Sharon Hutchinson, Helen M. Colhoun.

**Data curation:** Amanda Weir, Jen Bishop, Stuart J. McGurnaghan, Sharon Kennedy, David McAllister, Chris Robertson, Janet Murray, Alison Smith-Palmer, David Goldberg, Jim McMenamin, Colin Ramsay, Sharon Hutchinson.

**Formal analysis:** Paul M. McKeigue, Jen Bishop, Stuart J. McGurnaghan, Nazir Lone, Thomas M. Caparrotta, David Goldberg, Jim McMenamin, Colin Ramsay, Sharon Hutchinson, Helen M. Colhoun.

**Investigation:** Helen M. Colhoun.

**Methodology:** Paul M. McKeigue, Amanda Weir, Jen Bishop, Stuart J. McGurnaghan, Sharon Kennedy, David McAllister, Chris Robertson, Rachael Wood, Nazir Lone, Janet Murray, Thomas M. Caparrotta, Alison Smith-Palmer, David Goldberg, Jim McMenamin, Colin Ramsay, Sharon Hutchinson, Helen M. Colhoun.

**Resources:** Paul M. McKeigue, Amanda Weir, Sharon Kennedy, David McAllister, David Goldberg, Jim McMenamin, Helen M. Colhoun.

**Software:** Stuart J. McGurnaghan.

**Supervision:** David Goldberg, Helen M. Colhoun.

**Validation:** Paul M. McKeigue, Stuart J. McGurnaghan, Chris Robertson, Rachael Wood, Helen M. Colhoun.

**Visualization:** Paul M. McKeigue, Helen M. Colhoun.

**Writing – original draft:** Paul M. McKeigue, Rachael Wood, Nazir Lone, Janet Murray, Thomas M. Caparrotta, Alison Smith-Palmer, David Goldberg, Jim McMenamin, Colin Ramsay, Sharon Hutchinson, Helen M. Colhoun.

**Writing – review & editing:** Paul M. McKeigue, Amanda Weir, Jen Bishop, Stuart J. McGurnaghan, Sharon Kennedy, David McAllister, Chris Robertson, Rachael Wood, Nazir Lone, Janet Murray, Alison Smith-Palmer, David Goldberg, Jim McMenamin, Colin Ramsay, Sharon Hutchinson, Helen M. Colhoun.

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
