## [Editor Report · Decision Letter 0]

5 Jun 2020

Dear Dr Colhoun, 

Thank you for submitting your manuscript entitled "Rapid Epidemiological Analysis of Comorbidities and Treatments as risk factors for COVID-19 in Scotland (REACT-SCOT): a population-based case-control study" for consideration by PLOS Medicine.

Your manuscript has now been evaluated by the PLOS Medicine editorial staff [as well as by an academic editor with relevant expertise] and I am writing to let you know that we would like to send your submission out for external peer review.

Kind regards,

Clare Stone, PhD,

PLOS Medicine

---

## [Decision Letter · Decision Letter 1]

2 Jul 2020

Dear Dr. Colhoun,

Thank you very much for submitting your manuscript "Rapid Epidemiological Analysis of Comorbidities and Treatments as risk factors for COVID-19 in Scotland (REACT-SCOT): a population-based case-control study" (PMEDICINE-D-20-02483R1) for consideration at PLOS Medicine. 

[LINK]

In light of these reviews, I am afraid that we will not be able to accept the manuscript for publication in the journal in its current form, but we would like to consider a revised version that addresses the reviewers' and editors' comments. Obviously we cannot make any decision about publication until we have seen the revised manuscript and your response, and we plan to seek re-review by one or more of the reviewers. 

We expect to receive your revised manuscript by Jul 23 2020 11:59PM. Please email us (plosmedicine@plos.org) if you have any questions or concerns.

We look forward to receiving your revised manuscript. 

Sincerely,

Emma Veitch, PhD

PLOS Medicine

On behalf of Clare Stone, PhD, Acting Chief Editor,

PLOS Medicine

plosmedicine.org

*Please structure the abstract using the PLOS Medicine headings (Background, Methods and Findings, Conclusions - "Methods and Findings" should be a single subsection heading).

*In the last sentence of the Abstract Methods and Findings section, please include a brief note about any keyl imitation(s) of the study's methodology.

*At this stage, we ask that you include a short, non-technical Author Summary of your research to make findings accessible to a wide audience that includes both scientists and non-scientists. The Author Summary should immediately follow the Abstract in your revised manuscript. This text is subject to editorial change and should be distinct from the scientific abstract. Please see our author guidelines for more information: https://journals.plos.org/plosmedicine/s/revising-your-manuscript#loc-author-summary

*Did your study have a prospective protocol or analysis plan? Please state this (either way) early in the Methods section.

*The authors could consider using an appropriate reporting guideline (eg, RECORD - https://www.equator-network.org/reporting-guidelines/record/ - designed to support reporting of observational studies using routinely-collected data) to enhance reporting of methods and findings in their study. If the authors choose to use this please upload the completed RECORD (or other) checklist as supporting information with the revised paper.

Comments from the reviewers:

Reviewer #1: The authors used their unique linkage between administrative data holdings in Scotland to produce a case-control cohort of severely ill patients with COVID19, with general population controls (matched 1:7 on age, sex, and geography), documenting a series of risk factors for severe disease due to COVID19, potentially producing targeted populations for interventions in vulnerable cohorts. These validate previous reports from case series of severe disease, including prescription use, specific co-morbidities, age, and long-term care residence. 

Unique aspects of this approach are the population-level data to provide robust and standardized data on controls, aiming to produce a reasonable case-control series that provides inference on relationships between specific risk factors and presence of severe disease. 

Choice of matching by age/sex/geography is wise. Investigators have good experience with this dataset, and it appears cases and controls are reasonably well-matched in this regard. 

Conclusions are appropriately tempered, based on the data provided, suggesting that any number of comorbidities are associated with risk. 

Full case ascertainment of severe disease arguably limited by LTC deaths that do not reach hospital not accounted for, and may bias the rate ratios described.

Some further comments

- Challenges are the sheer number of univariate comparisons reported, and the limited corrections for multiple comparisons in place. Table 5 and 6, for example, are not particularly useful in interpreting the data, and arguably could be placed in the supplementary appendix, unless the purpose is to emphasize the non-specificity of the associations. 

-The presence of less robust associations in common conditions such as ischemic heart disease, and more robust associations in rare diseases like chronic kidney disease, is notable. This is mentioned in the discussion, but should be understood better, given the conclusions.

-A great deal of the Results should be moved to the Methods, such as the multivariable model construction, and the prescription association data. 

-Removing neoplasms from the prespecified list because of inability to discern who has already been shielded belies the fact that shielding is i) not 100% applied in populations; ii) not practiced in other regions, making the association between neoplasm and outcome an important one to consider for policy in other regions. 

-Abbreviation 'scrip' in Table 4 not universal, would use 'prescription'

-Very hesitant, given the sample size, the inaccurate labelling problem, the exclusive examination of one ethnic group, and the limited generalizability, to include any reporting or conclusions on the ethnicity-based risk factors. Would suggest excluding altogether, and reporting on the need for more robust ethnicity-based data collection strategies

-

Reviewer #2: See attachment

Michael Dewey

Reviewer #3: The article confirmed the previously reported or suspected risk factors, for example age, sex, underlying diseases, for severe COVID-19. The new findings were any previous admission to hospital, or any prescription and many comorbidities beyond those designated by public health agencies were also associated with severe COVID-19. Being a care home resident had the highest risk.

The authors defined severe COVID-19 as those who had entered a critical care unit or died within 28 days of first positive nuclei acid test. There were a total of 2755 severe cases. It would be informative if the authors could break down the severe cases into those that were due to death within 28 days without admission to critical care units, those that were admitted to critical care units and survived, those admitted to critical care but perished. Under normal circumstances, one would expect most of the death cases would be among the critical care unit cases. However, If a large proportion of the severe cases in the study were due to death within 28 days without critical care admission, one could still argue that those cases actually needed critical care, but were not admitted because of lack of critical care beds and subsequently perished, therefore were truly severe cases. However, one could also argue that a large portion of the severe COVID-19 cases were not "severe COVID-19 "cases and therefore, were not admitted to critical care unit, but died of other sudden events such as stroke, myocardial infection and so forth. Being a care home resident had the highest risk of severe cases, it would be interesting to analyze whether what proportion of severe cases among care home residents were death within 28 days without admission to critical care units. Providing the severe cases breakdown and a more in-depth discussion will further strengthen the conclusions. 

Table 3 showed association of Care Home, Any prescription, Any admission, and preexisting conditions with severe COVID-19. Since the authors concluded that many preexisting conditions were associated with severe COVID-19, may consider adding ANY comorbidity in the Table.

[LINK]

---

## [Decision Letter · Decision Letter 2]

31 Jul 2020

Dear Dr. Colhoun,

Thank you very much for re-submitting your manuscript "Rapid Epidemiological Analysis of Comorbidities and Treatments as risk factors for COVID-19 in Scotland (REACT-SCOT): a population-based case-control study" (PMEDICINE-D-20-02483R2) for review by PLOS Medicine.

I have discussed the paper with my colleagues and the academic editor and it was also seen again by one of the reviewers. I am pleased to say that provided the remaining editorial and production issues are dealt with we are planning to accept the paper for publication in the journal.

[LINK]

We look forward to receiving the revised manuscript by Aug 07 2020 11:59PM. 

Sincerely,

Clare Stone, PhD

Acting Chief Editor 

PLOS Medicine

plosmedicine.org

Requests from Editors:

Line 8 – Scottish ? national database? 

Abstract – please include p values (here and throughout) for quantifiable data and where 95% Cis are given. In addition, please add summary demographic information to the abstract and the study dates also need to be quoted. Please begin the "Conclusions" subsection of the abstract "In this study, we observed that ... were associated" or similar. 

Author Summary, please look at published examples, as your summary needs to be reformatted. It should be in bullet form with 3 headings. This text is subject to editorial change and should be distinct from the scientific abstract. Please see our author guidelines for more information: https://journals.plos.org/plosmedicine/s/revising-your-manuscript#loc-author-summary

Methods – the mention of the pre specified plan: Please provide a call out to where that can be accessed. 

Please remove “Declarations” on page 11 as these are pulled in automatically from EM in the metadata. 

Please use sections and paragraphs in reporting checklists as page number scan change in formatting and revisions. 

At line 400, is this study available as a preprint that can be cited, for example?

In the reference list, where you cite preprints please add "[preprint]".

Comments from Reviewers:

Reviewer #2: The authors have addressed my points.

Michael Dewey

[LINK]

---

## [Editor Report · Decision Letter 3]

18 Sep 2020

Dear Dr. Colhoun, 

On behalf of my colleagues and the academic editor, Dr. Srinivas Murthy, I am delighted to inform you that your manuscript entitled "Rapid Epidemiological Analysis of Comorbidities and Treatments as risk factors for COVID-19 in Scotland (REACT-SCOT): a population-based case-control study" (PMEDICINE-D-20-02483R3) has been accepted for publication in PLOS Medicine. 

PRODUCTION PROCESS

PRESS

PROFILE INFORMATION

Thank you again for submitting the manuscript to PLOS Medicine. We look forward to publishing it. 

Best wishes, 

Clare Stone, PhD

Managing Editor 

PLOS Medicine

plosmedicine.org